# A Causal Lens for Controllable Text Generation

**Zhiting Hu**[1,2], **Li Erran Li**[2]
[1]UC San Diego, [2]AWS AI, Amazon
zhh019@ucsd.edu, lilimam@amazon.com

## Abstract

Controllable text generation concerns two fundamental tasks of wide applications, namely generating text of given attributes (i.e., attribute-conditional generation), and minimally editing existing text to possess desired attributes (i.e., text attribute transfer). Extensive prior work has largely studied the two problems separately, and developed different conditional models which, however, are prone to producing biased text (e.g., various gender stereotypes). This paper proposes to formulate controllable text generation from a principled causal perspective which models the two tasks with a unified framework. A direct advantage of the causal formulation is the use of rich causality tools to mitigate generation biases and improve control. We treat the two tasks as *interventional* and *counterfactual* causal inference based on a structural causal model, respectively. We then apply the framework to the challenging practical setting where confounding factors (that induce spurious correlations) are observable only on a small fraction of data. Experiments show significant superiority of the causal approach over previous conditional models for improved control accuracy and reduced bias.

## 1 Introduction

Controllable text generation aims at producing fluent language with control over various attributes, ranging from sentiment, topic, politeness, to gender, persona, and so forth [60, 22]. The problem lies at the heart of many NLP applications such as emotional chatbot, news article writing, language detoxification, etc. Of particular interest in this increasingly significant area are two settings for control, namely (1) *attribute-conditional generation* [14, 30] which generates sentences that entail a given attribute, and (2) *text attribute transfer* [64, 26] which rewrites a given sentence to possess a desired attribute while preserving all other original characteristics (Figure 1). The goal is to learn the control in each setting with *(attribute, text)* training pairs[1].

The two settings have usually been considered as separate tasks and each led to various solutions, respectively. Let $x$ denote a sentence and $a$ an attribute. Previous attribute-conditional generation work typically concerns the conditional distribution $p(x|a)$ [14, 30, 33]. Despite the success of simulating observed real text, the conditional distribution is known to be susceptible to capture spurious correlations or biases in the data [48, 84]. For example, when generating biographical text given a gender attribute, the conditional model tends to generate text related to specific occupations such as nurse and yoga teacher for *female*, and architect and attorney for *male* [57, 69] (Figure 1). The learned biases could impair the model generalization to new domains, and make negative social impact in downstream applications. A few very recent attempts have been made to mitigate the biases in the model with various machine learning techniques. Yet those methods are often specific to a particular attribute (e.g., gender) [68, 86], or rely on access to additional resources, such as fully observed confounding labels or *a priori* debiased classifiers [23, 40, 67], which can be costly to obtain in real applications. Furthermore, it is unclear how the diverse methods designed for

---

[1]Thus, for text attribute transfer (a.k.a., text style transfer), there is no direct supervision data, i.e., *(original text, attribute, target text)* triples.

attribute-conditional generation could also be applied to debias text attribute transfer that has been formulated with distinct training objectives.

This paper studies controllable text generation from a principled causal perspective, that offers a unifying formulation of the two central tasks, and enables mitigation of spurious correlations with well-established causality techniques. A growing number of recent work has used causality with machine learning [62] for disentangled representation [54, 78], model explanation [6, 13], and robust prediction [61, 24, 83]. Yet most approaches have focused on the vision domain, taking advantage of image spatial structures, and thus are not directly applicable to text with abstract attributes (such as sentiment). Though previous research on text modeling has also studied related concepts such as counterfactuals, it either handles only correlation instead of causation [58, 39, 76, 23, 86, 46], or focuses on different applications such as data augmentation [85, 28, 82] and classification [13, 29]. We discuss more related work in §4.

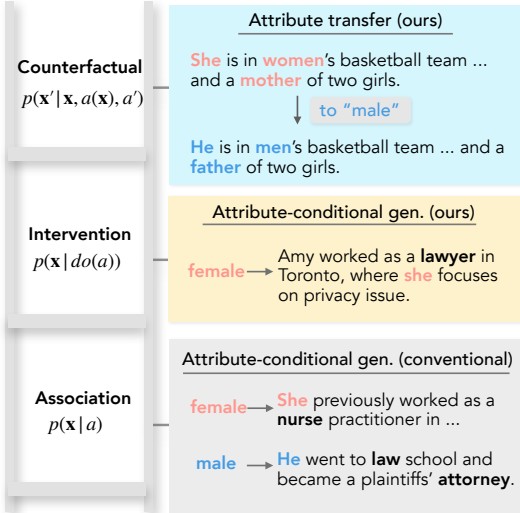

Figure 1: The causal ladder [56] and the formulations of controllable generation tasks corresponding to different rungs of the ladder.

We develop the first unified causal framework for text generation under control. In particular, we devise a structural causal model (SCM) [56] that describes the causal relationships between different variables, where the text $x$ is *outcome* and the attribute $a$ to control (e.g., sentiment) is *treatment*. The SCM further accounts for spurious correlations with *confounders* (e.g., category) with latent variables. The resulting SCM enables us to formulate the two control tasks as performing causal inference at different rungs of the causal ladder [56] (Figure 1), respectively. Specifically, (1) for attribute-conditional generation, we go beyond the association-based conditional $p(x|a)$ and propose to instead use $p(x|do(a))$, corresponding to the *intervention* rung. The $do$-operation effectively eliminates the effect of confounders on the control, leading to unbiased text outputs; (2) for text attribute transfer, the task naturally maps to *counterfactual* prediction on the SCM, which answers the question "what the text would have been if the attribute had been different" through the standard causal inference procedure [56, 55]. The unifying perspective also allows us to draw from existing successful techniques and train the SCM for accurate control and confounder balancing [27, 41, 22].

Previous causal work typically assumes access to confounding labels or relevant proxy information for the entire observed data [43, 44, 48]. In many real applications, however, it is prohibitively expensive or impossible to measure all the confounding factors for unbiased training. For example, it is often not affordable to annotate massively the confounding labels for the entire *(attribute, text)* corpus. We thus consider a more practical yet challenging scenario where we observe confounding information for only a small subset (e.g., $1\% - 5\%$) of samples [15]. We experiment on difficult datasets where the target attributes and confounding factors have strong correlations. Results show the causal approach substantially improves over conventional conditional models with enhanced control accuracy and reduced bias, on both attribute-conditional generation and attribute transfer.

## 2 Background

We first briefly review the causal concepts most relevant to the paper. A structural causal model (SCM) [56] is defined by a directed graph consisting of nodes (variables) and edges (direct causal dependence between variables), e.g., Figure 2. Different inference questions on an SCM correspond to different levels of the causal ladder (Figure 1) and require different reasoning tools: **(1)** "Association" deals with correlations in observed data with joint/marginal/conditional distributions. **(2)** "Intervention" concerns what would happen were some actions been performed. A typical question is to estimate the distribution of an *outcome* variable $x$ given an intervention on a *treatment* variable $a$: $p(x|do(a))$, where the $do$-operation represents an action on $a$ by setting it to a given value. With randomized experimental data (i.e., collected by randomly assigning treatment), $p(x|do(a))$ equals to the standard

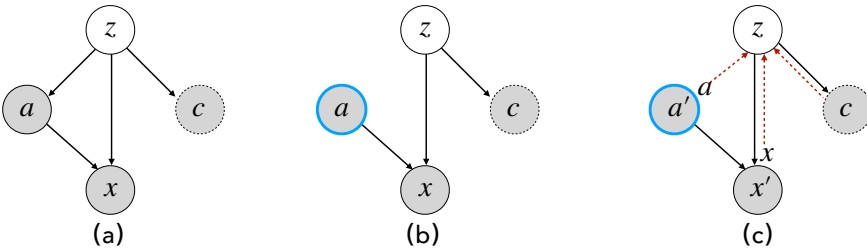

(a)  (b)  (c)

Figure 2: Illustration of causal graphs: **(a)** The proposed structural causal model (SCM, §3.1), where the outcome variable $x$ denotes the text, treatment variable $a$ denotes the attribute to control, $z$ is the latent confounder, and $c$ is the proxy variable for the confounder. A hollow circle indicates the variable is latent, and a shaded circle indicates the variable is observed. The proxy information $c$ is observed only for a subset of examples, which we indicate with a dashed circle. Note the difference of the SCM compared to previous latent-variable controllable generation models [22, 4] which do not explicitly model the confounder or its proxy information, making it impossible to identify the causal effects. **(b)** Intervention on the attribute $a$ (§3.2), represented as a blue circle, eliminates the dependence between $z$ and $a$, leading to the intervened SCM wherein the $z \to a$ arrow is removed. **(c)** Counterfactual prediction (§3.3), where red dashed arrows represent abduction from the original factual data $(a, x, c)$, and $x'$ is the counterfactual outcome given the new attribute $a'$.

conditional $p(x|a)$. Yet in practice, we usually only have access to passively observed data, such as the *(attribute, text)* pairs from existing corpus, and have to adjust for *confounders* (i.e., variables that correlate with both treatment and outcome) in order to estimate $p(x|do(a))$ from observational distributions. For example, we apply *backdoor adjustment* [56] in §3.2 for attribute-conditional generation. Finally, **(3)** "Counterfactuals" involves queries about what would have happened, given the knowledge of what in fact happened. We next show how the controllable text generation tasks are bridged together as different levels of causal inference, operationalized by the proposed SCM.

# 3 The Causal Framework for Controllable Text Generation

We now describe the unified causal perspective. We first develop the structural causal model that characterizes the causal structure in the controlled generation process (§3.1). We then show that intervention on the SCM leads to attribute-conditional generation (§3.2), while counterfactual prediction makes attribute transfer (§3.3). At last, §3.4 describes the training of the SCM with new objectives to encourage confounder balancing and de-correlation.

Compared to previous causal modeling in other domains (e.g., images), modeling text as the outcome is challenging due to the complex unstructured information encoded in the text. We show here that the unifying perspective enables us to bring to bear rich tools and inspirations from causal inference, disentangled representation, and controllable generation, for effective text causal modeling.

Figure 2 shows the SCM graphs as detailed below. In the appendix we illustrate the model architecture used in our experimental studies.

## 3.1 The Structural Causal Model

Figure 2(a) shows the SCM that describes the controlled generation process of text. Here the attribute of interest $a$ serves as the treatment, and the text $x$ is the outcome. For simplicity, we assume $a$ is binary (e.g., positive or negative sentiment), though the framework can straightforwardly be applied to more general cases where $a$ has multiple classes and dimensions. Note that $a$ as the condition for generating $x$ can be instantiated in different forms depending on the concrete application. For example, it can be a scalar $a \in \{0, 1\}$ as an input to the generator, or a word sequence such as $a \in \{$``[sentiment] positive''$,$ ``[sentiment] negative''$\}$ that acts as a prompt for the generator to produce text continuation [5, 18, 30, 81].

In general, the confounder that induces spurious correlations between $a$ and $x$ is infeasible to be fully specified or observed. For example, to control the sentiment of a restaurant review, the confounder could involve popularity of the restaurant, personal preferences of the customer, and other factors, whose values cannot be directly measured. Thus, following the recent causal approaches in other domains [43, 48, 44], we model the unobserved confounder as a high dimensional latent variable $z$, and infer $z$ from "indirect" confounding variables that are measurable in practice (such as food type).

The "indirect" variables are also called *proxy variables* in causality [50, 1, 70], which we denote as $c$. More background of confounder and proxy is provided in the appendix.

The goal of controllable text generation is thus to generate coherent text (given original text in the case of text attribute transfer) with accurate target attribute $a$ while unbiased in terms of the confounders.

Previous causal studies [43, 48, 44] have usually assumed the confounder proxy $c$ is available for all data. Similarly, recent work of debiasing attribute-conditional generation (based on machine learning) has relied on access to those extensive proxy labels [23, 40]. However, the assumption is often impractical due to the time and financial cost for obtaining the massive additional information beyond the common *(attribute, text)* data. We thus consider a more practical setting [15] where we only have access to the proxy information for a small subset (e.g., $1\% - 5\%$) of examples. In Figure 2(a), we use dashed circle of $c$ to denote the new challenging setting.

The resulting SCM thus defines a joint distribution

$$p_\theta(\boldsymbol{x}, a, \boldsymbol{z}, \boldsymbol{c}) = p_\theta(\boldsymbol{x}|a, \boldsymbol{z})p_\theta(a|\boldsymbol{z})p_\theta(\boldsymbol{c}|\boldsymbol{z})p_0(\boldsymbol{z}),  \tag{1}$$

where the component $p_\theta(\boldsymbol{c}|\boldsymbol{z})$ applies when the proxy $\boldsymbol{c}$ is observed for the example; $p_0(\boldsymbol{z})$ is a standard Gaussian prior following the common practice; and all components with free parameters $\boldsymbol{\theta}$ are modeled as deep neural networks. We use the amortized variational inference as in variational auto-encoders (VAEs) [31] to infer the latent confounder $\boldsymbol{z}$ from observations. Specifically, we introduce a variational distribution $q_\phi(\boldsymbol{z}|\boldsymbol{x}, a, \boldsymbol{c})$ with parameters $\phi$. To infer $\boldsymbol{z}$ for those examples whose proxy $\boldsymbol{c}$ is not available, we could apply an auxiliary predictor that estimates $\boldsymbol{c}$ from the observed $(\boldsymbol{x}, a)$. The auxiliary predictor can be trained on the subset of examples with available $\boldsymbol{c}$. In this work, we instead set the default $\boldsymbol{c}$ to a dummy value when inferring $\boldsymbol{z}$ for simplicity.

Note that previous work has also used VAEs for controllable text generation [22, 4]. However, they reside purely at the association level. In particular, despite the latent variables, they do not explicitly model the confounder and/or its proxy information, rendering the causal effects between components not *identifiable* [56]. As a result, those models are vulnerable to biases, as shown in the experiments.

### 3.2  Inference (I): Intervention for Attribute-Conditional Generation

We now discuss how to perform causal inference given the SCM for attribute-conditional generation. As mentioned in §1, in contrast to the conventional association-level methods based on the conditional $p(\boldsymbol{x}|a)$, here we formulate the task with the *interventional* conditional $p(\boldsymbol{x}|do(a))$. The $do$-operation sets $a$ to a given value independently of $\boldsymbol{z}$ (§2), which eliminates the dependence between $a$ and $\boldsymbol{z}$, leading to the new intervened causal graph in Figure 2(b), where the arrow from $\boldsymbol{z}$ to $a$ is removed. Thus $p_\theta(\boldsymbol{x}|do(a))$ captures the causal effect of attribute $a$ on text outcome $\boldsymbol{x}$ without confounding bias. We can use the *backdoor adjustment* [56] to estimate $p_\theta(\boldsymbol{x}|do(a))$ from the observed data:

$$p_\theta(\boldsymbol{x}|do(a)) = \sum_{\boldsymbol{z}} p_\theta(\boldsymbol{x}|a, \boldsymbol{z})p(\boldsymbol{z}).  \tag{2}$$

That is, we adjust for confounder $\boldsymbol{z}$ by making fair considerations of every possible $\boldsymbol{z}$ values and averaging the results by the distribution $p(\boldsymbol{z})$ discussed below. The difference from the previous methods becomes even clearer if we similarly decompose the conditional $p_\theta(\boldsymbol{x}|a) = \sum_{\boldsymbol{z}} p_\theta(\boldsymbol{x}|a, \boldsymbol{z})p_\theta(\boldsymbol{z}|a)$, which as we can see depends on $p_\theta(\boldsymbol{z}|a)$ and inherits the correlations between $a$ and $\boldsymbol{z}$ in the data.

We generate text samples from $p_\theta(\boldsymbol{x}|do(a))$ approximately by first drawing $\boldsymbol{z} \sim p(\boldsymbol{z})$ and then decoding with $\boldsymbol{x} \sim p_\theta(\boldsymbol{x}|a, \boldsymbol{z})$. To sample from the marginal $p(\boldsymbol{z})$ which does not have a clean analytic form, we use a similar approach as in [36] by fitting a simple generative adversarial network (GAN) [16], $p_{\text{GAN}}(\boldsymbol{z})$, on the learned latent space (i.e., $\boldsymbol{z} \sim q_\phi$ on all training data). We found it is sufficient to use a single-layer GAN which is fast to train.

### 3.3  Inference (II): Counterfactual for Text Attribute Transfer

Given an observed text $\boldsymbol{x}$, text attribute transfer seeks to produce new text $\boldsymbol{x}'$ that possesses the given new attribute $a'$ and preserves as many characteristics of the original $\boldsymbol{x}$ as possible. The task can naturally be mapped to counterfactual prediction on the SCM, i.e., imagining the alternative outcome for $\boldsymbol{x}$ should its attribute have been $a'$. The resulting inference procedure looks similar to the previous VAE-based attribute transfer method [22]. However, besides the key modeling difference of confounder/proxy as above, our counterfactual based interpretation offers a principled causal account

for the attribute transfer task. Moreover, the causal perspective inspires new training techniques that substantially improve the performance and reduce generation bias, as presented in §3.4.

Figure 2(c) illustrates the inference process. Specifically, from the causal perspective, counterfactual prediction is mathematically formulated as a three-step procedure [56, 55]: **(1) Abduction** that infers the "context" compatible with the observation $\boldsymbol{x}$. In our problem, it is sufficient to infer $\boldsymbol{z}$ as the context, as we would only intervene its descendant node $a$ in the SCM. Thus the step is done by computing $q_\phi(\boldsymbol{z}|\boldsymbol{x}, a, \boldsymbol{c})$; **(2) Action** that performs intervention on variable $a$ by setting $a = a'$; and **(3) Prediction** that computes the counterfactual outcome based on the SCM, i.e., $\boldsymbol{x}' \sim p_\theta(\boldsymbol{x}'|a', \boldsymbol{z})$, where we set $\boldsymbol{z}$ to the mean (vector) of the above abduction distribution $q_\phi$ for simplicity.

## 3.4 Learning

With the causal model and the inferences on it, we now discuss model training, which integrates variational learning and counterfactual reasoning for confounder balancing and disentanglement.

**Variational auto-encoding objective**    The base objective for learning the causal model is built on the common VAE approach [31]. Briefly, since the model's marginal log-likelihood (that marginalizes out the latent $\boldsymbol{z}$) is intractable, VAEs derive a lower bound with the variational distribution $q_\phi$. Formally, given a training example $(\boldsymbol{x}, a)$ with the optional proxy $\boldsymbol{c}$:

$$\mathcal{L}_{vae}(\boldsymbol{\theta}, \boldsymbol{\phi}) = \mathbb{E}_{\boldsymbol{z} \sim q_\phi} \left[ \log p_\theta(\boldsymbol{x}|a, \boldsymbol{z}) + \lambda_a \log p_\theta(a|\boldsymbol{z}) + \lambda_c \log p_\theta(\boldsymbol{c}|\boldsymbol{z}) \right] - \lambda_{kl} \mathrm{KL}\left(q_\phi \| p_0\right), \qquad (3)$$

where the first term is the reconstruction that aims to recover the observations $(\boldsymbol{x}, a, \boldsymbol{c})$ given the inferred $\boldsymbol{z}$ from $q_\phi$; the second term is a Kullback–Leibler regularizer that enforces the variational distribution to stay close to the prior $p_0(\boldsymbol{z})$. We refer readers to [31] for more details of VAEs. In the objective, $\lambda_a$, $\lambda_c$, and $\lambda_{kl} > 0$ are balancing hyperparameters. We set $\lambda_c$ to 0 when proxy $\boldsymbol{c}$ is not available, and otherwise select from $\{0.01, 0.1, 1\}$ based on validation, same as $\lambda_a$. We use the cyclic schedule from [36] to anneal $\lambda_{kl}$ from 0 to 1 to avoid excessive regularization of the KL term.

**Counterfactual objectives**    Training with the above base objective alone can lead to model collapse where the attribute variable $a$ is ignored in the generation process, i.e., text sampled from $p_\theta(\boldsymbol{x}|a, \boldsymbol{z})$ is not effectively controlled by $a$. This is because the training text $\boldsymbol{x}$ has already contained the attribute information, allowing both the inference of $\boldsymbol{z}$ and the subsequent reconstruction of $\boldsymbol{x}$ not to depend on the attribute value $a$. This issue highlights a key difference of our model compared to previous latent-confounder causal models in other domains (e.g., medication effect prediction), where the outcome is typically a simple binary variable (e.g., cured or not) that does not "leak" the treatment information (e.g., medication) [43, 48, 44]. Causal controllable text generation thus requires new solutions to encourage effective control.

Besides, a key ingredient for accurate causal inference is to achieve *balance* of confounders between treatment groups [27, 63, 52, 44]. That is, we want to match the confounder representation $\boldsymbol{z}$ of the examples whose $a = 0$ and those of the examples whose $a = 1$, in order to enhance the generalization performance for inferring counterfactual outcomes [27]. The concept is closely related to disentangled representation in machine learning which seeks to keep most dimensions of a representation invariant to the change of a particular dimension [41, 19].

The above two desiderata can be resolved with a suite of *counterfactual objectives* that are based on the counterfactual outcomes $\boldsymbol{x}'$ inferred in §3.3. We now describe those intuitive objectives, which are related to the attribute $a$, confounder $\boldsymbol{z}$, and proxy $\boldsymbol{c}$, respectively. We also discuss how we are able to draw inspirations from previous literature of disentangled representation and text attribute transfer, thanks to their connections with causal inference as above.

The **first objective** concerns the attribute $a$ to correctly learn its influence on the outcome. Intuitively, given the counterfactual outcome $\boldsymbol{x}'$ given the counterfactual attribute $a'$, we want to make sure $\boldsymbol{x}'$ truly entails $a'$. This can be achieved by using a pretrained attribute classifier $f(\boldsymbol{x}, a)$ that estimates the likelihood of text $\boldsymbol{x}$ possessing attribute $a$. More specifically, we train the model such that its predicted $\boldsymbol{x}'$ possesses $a'$ with a high likelihood measured by the classifier:

$$\mathcal{L}_{cf\text{-}a}(\boldsymbol{\theta}, \boldsymbol{\phi}) = \mathbb{E}_{\boldsymbol{z} \sim q_\phi, \, \boldsymbol{x}' \sim p_\theta(\boldsymbol{x}'|a', \boldsymbol{z})} \left[ f(\boldsymbol{x}', a') \right]. \qquad (4)$$

As in [22, 79], we use Gumbel-softmax approximation [47, 25] to the discrete text $\boldsymbol{x}'$ to enable gradient backpropagation for optimizing $(\boldsymbol{\theta}, \boldsymbol{\phi})$. Similar objective has been used in previous conditional generation of text [22, 45] and image [21, 35]. A crucial caveat is that, here the classifier itself

pretrained with the *(attribute, text)* data can also be biased due to confounding factors. Thus relying only on this objective as in the previous work is not sufficient for accurate unbiased attribute control, as shown in our experiments. To this end, we further devise the following counterfactual objectives.

The **second objective** focuses on balancing the confounder $z$. Intuitively, by the definition of counterfactuals, $x'$ must have the same confounder representation as the original $x$. We thus minimize the distance between the respective $z'$ and $z$:

$$\mathcal{L}_{cf\text{-}z}(\boldsymbol{\theta}, \boldsymbol{\phi}) = -\mathbb{E}_{\boldsymbol{z}, \boldsymbol{z}'}\left[d(\boldsymbol{z}', \boldsymbol{z})\right], \tag{5}$$

where, with slight abuse of notation, $z$ is the mean (vector) of $q_\phi(\boldsymbol{z}|\boldsymbol{x}, a, \boldsymbol{c})$, $z'$ is the mean (vector) of $q_\phi(\boldsymbol{z}'|\boldsymbol{x}', a', \boldsymbol{c})$ on the counterfactual $x'$, and $d(\cdot, \cdot)$ is a distance metric. Though the vectors $z$ and $z'$ have continuous values, we draw inspiration from the recent disentangled representation work [41] and use a binary cross-entropy loss to match $z'$ to $z$, i.e.,

$$d(\boldsymbol{z}', \boldsymbol{z}) = mean\left(\bar{\boldsymbol{z}}\log(\sigma(\boldsymbol{z}')) + (1 - \bar{\boldsymbol{z}})\log(1 - \sigma(\boldsymbol{z}'))\right), \tag{6}$$

where $\bar{\boldsymbol{z}} = (\boldsymbol{z} - \min(\boldsymbol{z}))/(\max(\boldsymbol{z}) - \min(\boldsymbol{z}))$) normalizes $z$ to $[0, 1]$, $\sigma(\cdot)$ is the logistic function applied to $z'$ element-wise, and $mean(\cdot)$ takes the average distance across $z$ dimensions. The distance is shown to be more effective [41] than the common $L_2$ loss $\|z' - z\|^2$ as used in earlier work [22].

The **third objective** carries similar intuition as above, though uses the proxy $c$ when it is available. Specifically, we want $z'$ to be able to reconstruct $c$ (as is $z$):

$$\mathcal{L}_{cf\text{-}c}(\boldsymbol{\theta}, \boldsymbol{\phi}) = \mathbb{E}_{\boldsymbol{z}'}\left[\log p_\theta(\boldsymbol{c}|\boldsymbol{z}')\right], \tag{7}$$

where $z'$ is the mean (vector) of $q_\phi(\boldsymbol{z}'|\boldsymbol{x}', a', \boldsymbol{c})$, same as in Eq.(5).

In sum, the overall objective for training the causal model is:

$$\mathcal{L}(\boldsymbol{\theta}, \boldsymbol{\phi}) = \mathcal{L}_{vae} + \gamma_a \mathcal{L}_{cf\text{-}a} + \gamma_z \mathcal{L}_{cf\text{-}z} + \gamma_c \mathcal{L}_{cf\text{-}c}, \tag{8}$$

with balancing hyperparameters $\gamma_a, \gamma_z$, and $\gamma_c \geq 0$. In practice, we found the model is not sensitive to the choices of those hyperparameters. We set each of them to either $0.5$ or $1.0$ based on validation.

## 4  Related Work

**Causal modeling for generation**    There is an emerging interest in integrating causality with machine learning [62] in various problems. Several latest works have studied causal inference combined with deep generative models for images, to learn causal structures between attributes [78, 65, 51], synthesize novel images [32, 3], and augment unbiased classifier training [61]. The spatial structure of images can make it easier to learn causal mechanisms, e.g., the work [61] specified independent modules for image background and texture. In contrast, text with abstract concepts (e.g., sentiment, topics) exhibits less independent structure. Previous causal modeling for text usually focuses on language understanding [29, 74, 7, 75, 71, 13, 49]. Recent work has also studied text as outcome in causal inference [12] for data augmentation [85, 28, 82] or generating text in specific domains (e.g., court view [77]). We make the first study of causal modeling for the general problem of text generation under control and demonstrate the effectiveness for bias mitigation.

**Controllable text generation**    Various approaches have been developed for attribute-conditional generation, by learning conditional language models (LMs) [30, 14, 81], guided inference [33, 9], or prompts [5, 72]. Recent work has focused on reducing gender bias in machine translation and generation [68, 66, 69, 11]. Other work studied more general unbiased generation with ML assuming access to unbiased classifiers [40, 23]. We use causal techniques to address a different and challenging setting where only limited confounding labels are observed. Unsupervised text attribute transfer has gained increasing attention [64, 22, 26], with the primary focus on learning to disentangle target attribute with other factors. We study the new challenge of attribute transfer in the presence of strong bias in the data, and show greatly improved performance.

## 5  Experiments

We study the challenging generation tasks with strong spurious correlations in training data. The causal framework substantially reduces bias and improves control accuracy.

We describe detailed model configurations in appendix. Briefly, the main model components, including the decoder $p_\theta(\boldsymbol{x}|a, \boldsymbol{z})$, inference network $q_\phi(\boldsymbol{z}|\boldsymbol{x}, a, \boldsymbol{c})$, and classifier $f(\boldsymbol{x}, a)$ (Eq.4) are all based on the GPT-2 (117M) architecture [59] with pretrained weights, respectively. In $q_\phi$ and $f$, we use the GPT-2 final-step output feature as the representation of input sentence $\boldsymbol{x}$. We implement other components ($p_\theta(a|\boldsymbol{z})$ and $p_\theta(\boldsymbol{c}|\boldsymbol{z})$) as simple MLPs. The model is trained with AdamW optimizer [42] using an initial learning rate of 1e-6. All experiments were conducted on 8 Tesla V100 GPUs.

## 5.1  Attribute-Conditional Generation

We first evaluate the interventional inference for attribute-conditional generation (§3.2). We use two datasets where the target attribute has a *correlation strength* of over 90% with the confounding factor, following the challenging settings of the latest work on visual bias [73, 17, 61, 65]. That is, the target attribute and the confounding factor of over 90% examples are both positive or negative, while those of the rest 10% examples are opposite. Differing from previous studies, we further assume the model can observe the confounding labels of a small subset of data, a more practical setting as in §3.1.

**Datasets**   Our **first** dataset is derived from the YELP challenge[2] that contains customer reviews of different categories. *Sentiment* (1:positive *vs.* 0:negative) is the attribute we aim to control, and the *category* of review object (1:restaurant *vs.* 0:others) is the confounding factor. Specifically, we extract a subset of data where 90% restaurant reviews are of positive sentiment, while 90% reviews to other entities (e.g., shopping) are of negative sentiment (thus a 90% correlation strength). We keep the category labels for less than 2% of training data. The resulting data has 510K/6K training/validation examples, wherein 10K training examples have observable confounding category labels[3]. For evaluation, we further create a balanced test set of 13K examples with correlation strength 50% (i.e., no correlation). Following the previous controllable generation [22, 64], we focus on generating short text, by truncating the output text in the data to 20 tokens at maximum.

The **second** dataset is from the BIOS corpus [10] that contains online biographies with gender and occupation labels. We use gender (female/male in the corpus) as the attribute to control. Thus the goal is to generate biographical text of a given gender. For occupation which is the confounding factor, we subsample and merge the occupations into two groups, i.e., *{nurse, dietitian, paralegal, ... }* and *{rapper, DJ, surgeon, ... }* (see appendix for more details). The correlation strength of the resulting dataset is 95%. For example, 95% female biographies are about the occupations in group one. We randomly split the dataset into 43K training and 2K validation examples, and keep the binary occupation labels for only 3K randomly selected training examples (among which only 5%×3K=150 examples have opposite gender and occupation labels). As above, we further create a balanced test set of 2K examples for evaluation, and truncate the output text to no more than 20 tokens.

**Baselines and setup**   We compare with the conditional language models that people would commonly train for the task. The first model, `Conditional LM`, conditions only on the target attribute and generates text accordingly. The second model, `Conditional LM (full)`, makes full use of the attribute and confounding labels in hope of better de-correlating the two. Since the confounding labels are available only on a small subset of examples, we first train a classifier on the subset with data-reweighting (see appendix for details), and use it to predict confounding labels for the remaining examples. The language model is then trained on the resulting complete data, conditioning on both the attribute and the (real or estimated) confounding label. We also compare with latest attribute-conditional generation approaches, such as `GeDi` [33] where a language model conditioning on the confounding information $p_{\text{gedi}}(\boldsymbol{x}|\boldsymbol{c})$ is used to reshape the generation distribution of the above `Conditional LM`. We include comparison with more baseline methods in the appendix.

For our approach, the available confounding labels serve as the proxy $\boldsymbol{c}$. The attribute classifier $f$ used to train our model (Eq.4) is pretrained on the biased training data. On YELP, the resulting (sentiment) classifier has a mediocre accuracy of 83% on the balanced test set; On BIOS, the (gender) classifier has an accuracy of 91%.

---

[2]https://www.yelp.com/dataset/challenge

[3]Due to the strong correlation, only 10%×10K=1K examples have opposite sentiment and category labels, posing a significant challenge for the model to de-correlate the two factors.

|  | Methods | Control accuracy (↑) | Bias (↓) | Fluency (↓) | Diversity (↑) |
|---|---|---|---|---|---|
| YELP | Conditional LM | 79.1 | 78.7 | 50.4 | 41.4 |
| | Conditional LM (full) | 80.3 | 78.9 | 50.8 | 41.9 |
| | GeDi [33] | 80.9 | 74.3 | 83.2 | 41.7 |
| | Ablation: Ours w/o $cf$-$z/c$ | 91.1 | 89.2 | 54.1 | 40.4 |
| | Ours | **96.3** | **59.8** | 51.3 | 39.1 |
| BIOS | Conditional LM | 95.51 | 84.73 | **17.0** | 46.5 |
| | Conditional LM (full) | 93.28 | 72.34 | 18.5 | 48.5 |
| | GeDi [33] | 86.0 | 75.2 | 27.8 | 43.5 |
| | Ablation: Ours w/o $cf$-$z/c$ | 97.3 | 70.1 | 29.4 | 42.1 |
| | Ours | **99.2** | **62.4** | 32.0 | 40.6 |

Table 1: Automatic evaluation of attribute-conditional generation on YELP and BIOS. *Control accuracy* is measured by the attribute classifier accuracy; *Bias* is by the confounding classifier accuracy; For *fluency*, we report perplexity, thus a lower score indicates more fluent text; *Diversity* is measured by the Distinct-2 metric. For each evaluation aspect, we highlight the best result that has significant improvements over others.

|  | Methods | Control accuracy (↑) | Bias (↓) | Fluency (↑) |
|---|---|---|---|---|
| YELP | Conditional LM (full) | 80.0 | 73.0 | 3.90 |
| | Ours | **97.0** | **56.0** | 3.85 |
| BIOS | Conditional LM (full) | 96.0 | 82.0 | 4.43 |
| | Ours | **99.0** | **60.0** | 4.25 |

Table 2: Human evaluation of attribute-conditional generation on YELP and BIOS.

**Evaluation**   We conduct both automatic and human evaluation. For the former, we follow the common practice and evaluate the generations in terms of various aspects as following: **(1) Control accuracy** for which we use an "evaluation attribute classifier" that takes as inputs the generated sentences and measures how accurate they entail the input attributes. The evaluation attribute classifier is trained on a large unbiased set of examples from the original corpus and is of high test accuracy (87% for YELP and 95% on BIOS) for evaluation purpose (note the difference from the above classifier $f$ trained with only biased training data); **(2) Bias** which is measured by another classifier for the confounding factor. Intuitively, the better the predicted confounding labels match the input attributes, the more correlated the two factors in the generation. A 50% match indicates no correlation. The classifiers are trained similarly as the evaluation attribute classifiers, and achieve accuracy 85% on YELP and 90% for BIOS; **(3) Fluency** which is measured by applying GPT-2 language models (LMs) on the generated text and computing the perplexity; The LMs obtain perplexity of 32.4 and 18.0 on the real text of YELP and BIOS, respectively. **(4) Diversity** with the common Distinct-$n$ metric [37] that measures the ratio of unique $n$-grams against total number of $n$-gram in the generation set. We evaluate 10K generated samples by each model.

For human evaluation, we ask human raters to annotate for each generated text the attribute label and confounding factor label, based on which we compute the control accuracy and bias as above. We also annotate language fluency using a 5-point Likert scale. On each dataset, we compare `Conditional LM (full)` and our approach, with 100 sentences from each model annotated by 3 raters. The Pearson correlation coefficient of human scores is 0.67, showing strong inter-rater agreement.

**Results**   Table 1 shows the automatic evaluation results on both YELP and BIOS. Our causal approach significantly improves over the association-based conditional models. For example, on YELP, our model achieves 16% absolute improvement in terms of control accuracy, and at the same time reduces the bias (spurious correlation with the confounder) by 19%. In contrast, the conditional LMs mostly inherit the bias from the training data. As an ablation study, we also evaluate a simplified variant of our full approach by omitting the counterfactual objectives w.r.t $z$ and $c$ (Eqs.5 and 8) (which reduces to a training strategy similar to the previous methods [e.g., 22]). The variant improves the control accuracy over the conditional LMs, but fails to effectively reduce the generation bias. The results show the crucial role of confounder balancing in bias reduction. On the BIOS dataset, our approach also obtains consistent improvement on both accuracy and bias.

Table 2 shows the human evaluation results on both datasets, which largely confirm the above observations with automatic evaluation.

| Methods | Control accuracy ($\uparrow$) | Bias ($\downarrow$) | Preservation ($\uparrow$) | Fluency ($\downarrow$) |
|---|---|---|---|---|
| Hu et al. [22] | 44.1 | 68.4 | 77.7 | 132.7 |
| He et al. [20] | 35.3 | 60.2 | **80.1** | 57.7 |
| Ablation: Ours w/o $cf$-$z/c$ | 75.0 | 67.8 | 36.3 | 34.2 |
| Ours | **77.0** | 61.4 | 42.3 | **29.6** |

Table 3: Results of attribute transfer on the *biased* YELP. Baselines [22, 20] with the public code, and they fail to rewrite the text on most instances, leading to very low control accuracy and high preservation.

| Methods | Control accuracy ($\uparrow$) | Preservation ($\uparrow$) | | Fluency ($\downarrow$) |
|---|---|---|---|---|
| | | self-BLEU | ref-BLEU | |
| Hu et al. [22] | 86.7 | **58.4** | - | 177.7 |
| Shen et al. [64] | 73.9 | 20.7 | 7.8 | 72.0 |
| He et al. [20] | 87.9 | 48.4 | 18.7 | **31.7** |
| Dai et al. [8] | 87.7 | 54.9 | 20.3 | 73.0 |
| Ablation: Ours w/o $cf$-$z/c$ | 87.1 | 57.2 | 24.3 | 46.6 |
| Ours | **91.9** | 57.3 | **25.5** | 47.1 |

Table 4: Results of text attribute transfer on the common *unbiased* YELP data.

## 5.2 Text Attribute Transfer

We next study text attribute transfer (§3.3) as the second core task of controllable generation. The proposed causal approach also achieves substantial improvement in terms of accurate control and bias reduction. Besides, for a broader comparison, we also apply our approach to another unbiased dataset widely studied in previous text attribute transfer research, showing superior performance.

**Datasets** We use the above biased YELP dataset (§5.1) to study the attribute transfer, where we aim to modify a sentence to possess the opposite sentiment (e.g., from negative to positive), and at the same time preserve all other characteristics. In particular, we want the new sentence to keep the category unchanged, which is difficult for previous association-based controllable models given the strong correlation in the data between sentiment and category. Besides, since most previous attribute transfer studies have focused only on unbiased setting, we additionally evaluate our approach on the popular *unbiased* YELP data (reviews with sentiment for restaurants only) [64] for comparison.

**Evaluation** We follow the standard practice for evaluation. For the biased setting, we measure **control accuracy**, **bias**, and **fluency** as in §5.1. We also assess the common aspect **preservation**, which evaluates the BLEU score [53] between the generated and original sentences (i.e., self-BLEU). A higher score indicates better preservation of sentence properties. For the unbiased setting, we omit the bias evaluation, and additionally compute another preservation metric, ref-BLEU, which is the BLEU score between the generation and human-written golden text on a subset of test examples [38]. We also conduct **human evaluation** which shows the same conclusions as the automatic evaluation in terms of model performance. We put the results in appendix due to space limitation.

**Results** Table 3 shows results on the biased YELP data, a substantially more challenging setting than the popular unbiased one (Table 4). We compare two of the previous best-performing methods with public code. Our approach again manages to reduce the bias while achieving decent transfer accuracy. The previous methods struggle to edit the text on many instances (e.g., generating the same sentences as inputs), leading to low control accuracy. Ablation comparison with our simplified variant (our `w/o cf-z/c`) further validates the effect of counterfactual objectives for confounder balancing (§3.4), as shown by the improved accuracy and mitigated bias of the full approach.

Finally, Table 4 shows the results on the common unbiased YELP sentiment data. The results show our approach generates fluent output with improved accuracy and preservation.

## 6 Conclusions and Future Work

We have presented a principled causal perspective for the two core tasks of controllable text generation. Based on the proposed structural causal model, attribute-conditional generation is modeled as interventional inference, and text attribute transfer performs counterfactual prediction. We connect

rich techniques in causality, disentangled representation, and text generative modeling, and develop learning objectives for accurate control and confounder balancing. Focusing on the challenging setting with partially available confounding information, the experiments show our approach achieves accurate control and mitigates the strong correlations in the data.

The proposed causal framework opens up a range of new opportunities for further improving and enriching controllable text generation. For example, though this work has focused on single control attribute and confounding factor, it would be interesting to generalize the approach for structured control of a richer set of text attributes, by modeling the underlying **causal graph between attributes** (as explored similarly in image generation [78, 65]). Besides, we are interested in importing more causality tools through the causal perspective to enable new applications. For instance, the *inverse propensity reweighting* technique in causality can potentially be used to **debias pretrained language models** $p_{\text{pretrain}}(\boldsymbol{x}|a)$, with the following known equation between the unbiased interventional conditional $p(\boldsymbol{x}|do(a))$ and the biased standard conditional $p(\boldsymbol{x}|a)$:

$$p(\boldsymbol{x}|do(a)) = \sum_{\boldsymbol{z}} p(\boldsymbol{x}|a, \boldsymbol{z})p(\boldsymbol{z}) = \sum_{\boldsymbol{z}} p(\boldsymbol{x}|a)p(\boldsymbol{z}|\boldsymbol{x}, a)\frac{p(a)}{p(a|\boldsymbol{z})}, \tag{9}$$

where $p(a|\boldsymbol{z})$ is known as the *propensity score* [56], i.e., the propensity (probability) of the $\boldsymbol{z}$ being assigned to the particular treatment $a$. Plugging in the $p_{\text{pretrain}}(\boldsymbol{x}|a)$ together with the parameterized estimates of $p_\theta(\boldsymbol{z}|\boldsymbol{x}, a)$ and $p_\theta(a|\boldsymbol{z})$ as learned in §3, we would effectively convert the pretrained LM into the unbiased $p(\boldsymbol{x}|do(a))$. Further, rich studies in the causality literature have proposed stabilized and enhanced variants of the above inverse propensity reweighting [e.g., see 80], all of which present interesting topics to explore in the controllable generation setting in the future.

**Ethical considerations**   We would like to note that automatic text generation could be used maliciously to generate fake, toxic, or offensive content [34, 72, 2]. We hope the unbiased modeling study could offer techniques to alleviate potential issues.

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
