# OpenReview forum: "A Causal Lens for Controllable Text Generation"
_NeurIPS.cc/2021/Conference — NeurIPS 2021 Poster_

### Official Review · Reviewer_dfNU · 2021-07-09

**Rating:** 7
**Confidence:** 3

**Summary:**

The goal of the paper is to control an attribute of the text without changing another associated attribute. Let's use the biased YELP dataset the authors created as an exampe. In the dataset, 90% of positive reviews come from restaurants and 90% of negative reviews come from the other categories (e.g., shopping). Their goal is to control the sentiment (attribute a) without changing its category (confounding factor c). This is difficult because the conditional generator might just generate a restaurant review when the task is to generate a positive review. Similarly, if most of the nurses are female, the text generator might just generate nurses to satisfy the requirement of generating a female character.

In conditional generation such as a seq2seq model, even the controlled attribute is given as the condition, the text generator might still generate the text that might imply confounding factors or some other attributes because the confounding factors and controlled attribute are associated or the prior probabilities of those factors/attributes are high. The paper solves the problem using backdoor adjustment (removing the dependency between the attribute a and the hidden representation of the sentence z). My understanding of their method is like assuming confounding factors and controlled attributes are independent or ignore the prior probability of observing other attributes. This would make the generated text less fluent but can generate the text that would have the attribute in a high probability.

The experiments show that the proposed method is significantly better than conditional LM or other text attribute transferring approaches when the training data is biased (e.g., the review category is highly correlated with the sentiment of the review in YELP). When the dataset is unbias, one experiment seems to show that the proposed method is also better than other approaches.


**Limitations And Societal Impact:**

One potential negative societal: Although neural style transferring could be used to remove toxic attributes but also could be used to inject toxic attributes, I do not think explicitly mentioning this possibility would have a positive social impact. Thus, I think the paper should keep as it is on this part.

**Main Review:**

I think the problem setting is practical. For example, people might want to remove an undesired attribute that has a high correlation with other desired attributes (e.g., remove the toxic attribute but keep the negative sentiment). The proposed method also seems to work pretty well. I did not know any other direct-related work. Thus, the originality and significance are good, which might lead to high impact.

Nevertheless, the experiments and presentation still have sufficient room for improvement. First, the paper does not say very clearly whether the proposed method is better than existing approaches in the popular unbiased benchmarks. The evaluation setting of the unbiased YELP experiment is not specified clearly. Is the proposed model trained on the unbiased dataset in this experiment? How do you tune the hyperparameters of your model and other methods? Do you think where the improvement comes from? Do all the models use the same amount of resources and are trained on the same data? For example, does the proposed method use category information but other methods do not? Is your model size (number of parameters) larger than previous approaches (I suggest you put all the model sizes into your table)? Is the classifier given when you train other models? Do all the models use the same classifier? Could you also compare the diversity in the text attribute transfer task? Could you also compare your methods with other approaches in a more standard evaluation (e.g., Table 9 in https://arxiv.org/pdf/2010.12742.pdf)? You use the predicted confounding labels to train Conditional LM (full), so I am not sure whether it performs worse than your model due to the noise in the predicted labels or the effectiveness of your model. Did you try to train your casual model on such predicted labels rather than the ground truth label subsets?

If the authors cannot show that the proposed method is consistently better than other approaches in unbiased datasets in a fair comparison. I think emphasizing the focus of this paper earlier would make the paper much easier to read. For example, the current title and introduction give me the impression that this paper proposes a general approach to all types of controllable text generation tasks. I do not understand the main motivation behind the paper until I saw the creation of the biased YELP dataset in the experiment section. I recommend the author to mention some concrete examples (e.g., in your biased YELP dataset) in the introduction or Figure 2 to explain what a, c, x, and z mean and emphasize the focus of this paper. In addition, in Tables 1 and 2, the YELP dataset should be called biased YELP to avoid confusion.

Finally, I think more ablation studies are needed (The paper only tries another baseline in the ablation study). The architecture is complicated (e.g., line 162 says it even uses GAN) and you have seven loss terms. Why do we need to use GAN here? If the proposed method is better than other approaches in the unbiased setting, we would like to know how much each model design or each loss term contributes to the improvement.

If the rebuttal addresses my concerns about experiments and presentations well, I will vote acceptance (>=7) for this work. You don't have to answer all the questions I asked. I just want to know whether this method is generally better than state-of-the-art results in unbiased datasets in a fair comparison. If no, majorly revise your presentation. If yes, provide more pieces of evidence (and minorly revise the paper to make your motivation more concrete).


Some minor suggestions:
1. More detailed description of model architectures should be included in the appendix. It would be even better to add a figure to explain architecture. This is especially important if you want to claim the proposed method is generally better than others in unbiased datasets. I only have a very high-level understanding of the model after reading the whole paper.
2. A paper I saw usually won't describe the baselines inside the result section/paragraph. Explain cf-z/c baseline before the result paragraph and mention the cf-z/c keywords that allow the readers to search.
3. In the paper, you mentioned that your method is related to disentangled representation studies. The relation of your work to the related text style transfer studies such as [1] could be discussed in the paper.
4. I think the perplexity numbers are usually positive values and smaller is better.
5. What's the n value you used in the distinct-n diversity metric?

[1] John, Vineet, et al. "Disentangled Representation Learning for Non-Parallel Text Style Transfer." Proceedings of the 57th Annual Meeting of the Association for Computational Linguistics. 2019.

**Time Spent Reviewing:**

5

---

> ### Author Response · Authors · 2021-08-10
> **Response**
>
> Thank you for your positive and insightful comments.
>
> &nbsp;
>
> **Main focus of the paper**
>
> We’d like to first clarify that the main focus of the paper is a unified causal framework for controllable text generation, which has the *key advantage* of reducing bias (spurious correlations) in data with the principled causality techniques. And thanks to the unified framework, the same techniques can be applied to debias different generation tasks including attribute-conditional generation and text style transfer.
>
> Accordingly, our experiments were primarily designed to validate the key advantage of the framework, i.e., learning unbiased controllable generation from biased datasets -- a common and crucial problem in practice. As discussed in Lines.33-39, recent work has started to study the debiasing problem especially for attribute-conditional generation, but they have limitations (e.g., specific to the “gender” attribute, and/or requiring additional resources), and are not easily generalizable to the task of text attribute transfer.
>
> We’d like to note that we’ve highlighted the “debiasing” idea in Abstract (Lines.5-6, Line.8) and Introduction (Lines.28-39, and Figure.1). We’ll revise both the title and the writing to make the key message clearer. Thanks for the suggestion!
>
> &nbsp;
>
> **Setting for the unbiased YELP experiment**
>
> Though we intend to focus on the biased generation setting, we also show the comparison (Table.4) of the proposed approach v.s. previous models on the unbiased dataset (as it’s the focus in previous studies). We answer your questions below. We’ll polish the paper to make them clearer.
>
> - The proposed model is trained on the unbiased dataset in this experiment. That is, all comparison models are trained on the same data. The proposed model does not use any category information as it’s not available in the unbiased YELP dataset. (The proposed method is flexibly applicable to settings with no category information, with partial category information, or with full category information.)
>
> - We copied the results of baselines from previous papers. We selected the hyperparameters of our method as discussed in section 3.4 (Lines.190-192, 238-239)
>
> - Regarding “where the improvement comes from”, we’ve reported the results of *“Ours w/o cf-z/c”* for ablation. As noted in Lines.334-336, the training of *“Ours w/o cf-z/c”* is similar to the baseline Hu et al., [20]. The key difference is the model architecture (e.g., we used GPT-2 as decoder). Thus, comparing *“Ours w/o cf-z/c”* with [20] shows the effect of the model architecture (i.e., improved fluency). Comparing *“Ours w/o cf-z/c”* with our full model shows the effect of the proposed counterfactual training objectives (i.e., improved control accuracy and preservation) -- the counterfactual objectives (in particular Eqs.5 & 6) effectively encourage attribute disentanglement.
>
> - We use BERT/GPT-2 in our model (including the classifier), which has a larger model size than previous models (which typically use LSTM). We’ll include the model size in the table. The above ablation shows the effect of the different architecture.
>
> - We followed the survey paper [24] (e.g., its Table.5) for standard evaluation of the task, which is similar to that of https://arxiv.org/pdf/2010.12742.pdf you pointed out. We’ll report the additional metrics mentioned in the paper as you suggested. Note that “diversity” is not suitable for text attribute transfer by definition, since the task requires preserving the source content as much as possible. (Both survey papers do not include diversity as an evaluation aspect).
>
>
> &nbsp;
>
> **Real and predicted confounding labels in training**
>
> As noted in Lines.304-305, the *Conditional LM (full)* baseline uses both the real and predicted confounding labels in training. Our model uses exact the same amount of real confounding labels. (Our model does not require explicit confounding label prediction on other data instances whose real labels are unavailable, which is an advantage over the baseline). Thus, the comparison is fair, and the improved performance is due to the effectiveness of our model.
>
> &nbsp;
>
> **GAN in our method**
>
> GAN is introduced to estimate the marginal latent distribution `p(z)` in backdoor adjustment (Eq.2). Note that the GAN is of minimal size (single-layer MLP) and very easy/fast to train. The similar approach has been used in previous VAE work [33].
>
> &nbsp;
>
> **Complicated loss terms; ablation studies**
>
> Though the full method includes seven loss terms, the first four terms shown in Eq.(3) are just the standard VAE objective. Our method introduces the additional three counterfactual terms to avoid model collapse and encourage debiasing. Among the three, the first counterfactual term (Eq.4) is indispensable to prevent model collapse (discussed in Lines.193-201). To better show the effect of the remaining two counterfactual terms, we conduct additional ablation study -- please see our response to R2 (Reviewer BdYj) for the results and discussion.
>
> &nbsp;
>
> **Other minor questions**
>
> - We’ll add a figure to illustrate the model architecture and add more descriptions. (Briefly, as discussed in appendix B, our model architecture follows the previous work [6] with BERT as the encoder and GPT-2 as the decoder).
> - We’ll revise the description of the ablation models for clarity.
> - We’ll add more discussion of related text style transfer studies as you suggested.
> - We use n=2 in the Distinct-n diversity metric (see the caption of Table.1). We’ll make it clearer.
>
> We’ll fix all other minor issues. Thanks for your suggestions!

---

> > ### Comment · Reviewer_dfNU · 2021-08-10
> > **Response addresses most of my concerns**
> >
> > I have updated my score to 7.
> >
> > Since you are using pre-trained GPT-2/BERT model while the baselines use LSTM, I think you should not claim or imply that the bias correction part in your model is helpful to the unbiased dataset. I hope that you could majorly revise your presentation to let the readers see the concrete examples this paper focuses on as early as possible.
> >
> > Finally, I encourage the authors to further verify that the claim that the proposed model is doing better than Conditional LM (full). A machine learning model usually performs worse when it is trained on the predicted label due to error propagation. If your model also does not work well after training on both real and predicted labels, it is very likely that Conditional LM (full) did worse simply due to the error propagation. Another thing you can try is to first pretrained Conditional LM (full) without inputting confounding labels on all the data and then fine-tune the Conditional LM (full) on the small subset of data with the real labels and see how well it performs.

---

> > > ### Author Response · Authors · 2021-08-22
> > > **Response to new comments**
> > >
> > > Thank you for updating the score, and thanks for the suggestions!
> > >
> > > &nbsp;
> > >
> > > **Presentation**
> > >
> > > We’ll update our presentation and add more concrete examples in the beginning of the paper. Thanks for the suggestion! In the unbiased data setting, as mentioned in our response above, the effect of the proposed causality-based learning/inference (instead of the model architecture) is shown by the comparison with the ablation model *“Ours w/o cf-z/c”*. We’ll make this clear to avoid confusion as suggested.
> > >
> > > &nbsp;
> > >
> > > **Conditional LM (full)**
> > >
> > > Augmenting the training of *Conditional LM (full)* with predicted labels (though noisy) is to an extent helpful. To show this, below we report the results of the baseline trained in the way as you suggested --- i.e., we first pretrain *Conditional LM (full)* without inputting confounding labels on all the data, and then fine-tune the *Conditional LM (full)* on the small subset of data with the *real* confounding labels.
> > >
> > > From the able below, we can see that, fine-tuning on only the small subset of data with real labels, the *Conditional LM (full)* obtains a low control accuracy (74.1). Our method outperforms both versions of *Conitional LM (full)* (i.e., with or without predicted labels) in terms of both the control accuracy and reduced bias. We’ll add more results and discussion.
> > >
> > > &nbsp;
> > >
> > > *More results of Conditional LM (full) on biased Yelp:*
> > >
> > > | Methods   | Control accuracy  | Bias (the lower the better)  |  Fluency  |  Diversity  |
> > > |---|---|---|---|---|
> > > | Conditional LM (full) (reported in Table.1)  | 80.3  | 78.9  | -50.8  | 41.9 |
> > > | Conditional LM (full) (real labels only)  | 74.1  | 64.3  | -47.1  | 40.9  |
> > > | Ours (reported in Table.1) | **96.3**  | **59.8**  | -51.3  | 39.1 |

---

> > > > ### Comment · Reviewer_dfNU · 2021-08-22
> > > > **Thanks for the new results**
> > > >
> > > > Thank you for taking my advice.
> > > > It's good to see that the conditional LM in your paper is indeed strong.
> > > > I have no further questions or doubts.

---

### Official Review · Reviewer_nG9H · 2021-07-13

**Rating:** 6
**Confidence:** 4

**Summary:**

This paper investigates the problem of the attribute (sentiment, gender) controlled text generation or text style transfer. The authors formulate the problem from a principled causal perspective, which models the two tasks with a unified framework. A framework of structural causal model is introduced to ground the idea. Experiments on text generation and style transfer are conducted and the experimental results show that the proposed framework can indeed improve the control accuracy.

**Limitations And Societal Impact:**

The authors have addressed the limitations and potential negative societal impact.

**Main Review:**

This paper investigates the problem of the attribute (sentiment, gender) controlled text generation or text style transfer. The authors formulate the problem from a principled causal perspective, which models the two tasks with a unified framework. A framework of structural causal model is introduced to ground the idea. Experiments on text generation and style transfer are conducted and the experimental results show that the proposed framework can indeed improve the control accuracy.


Strength:

- The paper is well written and easy to follow. The authors describe the technical content detailly and readers can easily understand the implementation tricks.

- The proposed framework unifies the tasks of attribute-controlled text generation and style transfer which using one model can conduct two tasks simultaneously.


Weakness:

- Although the authors present the story from a principled causal perspective which is good certainly, it is not easy to figure out the novelty and technical contribution compared with the previous methods (such as CVAE model). The basic and main idea is still a VAE-based framework with some more variables or tricks to improve the performance, therefore, from the technical derivatives and descriptions, it is hard to feel excited and benefit more.

- It is good to see that Conditional LM is introduced as the baseline method to conduct comparisons. I am wondering what is the performance of CVAE model which GPT2 is employed as the language decoder. This comparison is more important and convincing.



**Time Spent Reviewing:**

2

---

> ### Author Response · Authors · 2021-08-10
> **Response**
>
>
> **Technical novelty**
>
> The proposed causal framework is fundamentally different from CVAE and other machine learning methods which are at the association level in the Causal Ladder (Figure.1) and susceptible to bias in the data. The principled causal formulation allows us to apply rich causality tools to **debias** the controllable generation, such as the *backdoor adjustment* in Eq.(2) for inference, which reduces the spurious correlation between the target attribute and the confounder, and the counterfactual objectives (section 3.4) inspired by confounder balancing for training, which further remove the attribute-confounder dependencies and reduce bias. That is, the proposed framework has very different inference and training methods than the standard VAE.
>
> &nbsp;
>
> **Comparison with CVAE**
>
> As above, CVAE as an association-level method is not able to remove the bias in data due to confounders. To show that, we add the comparison with CVAE. Specifically, we make both the target attribute (sentiment) and confounder (category) as the conditions for CVAE, similar to the *Conditional LM (full)* baseline in the paper. We use the same model architecture (e.g., GPT2 as decoder) for a fair comparison. The results are reported below. As expected, we can see that CVAE is close to *Conditional LM (full)*, both of which inherit the high degree of bias from the data. In contrast, our method with the above-mentioned new training/inference techniques, managed to improve the control accuracy and greatly reduce the bias.
>
> &nbsp;
>
> | Methods   | Control accuracy  | Bias (the lower the better)  |  Fluency  |  Diversity  |
> |---|---|---|---|---|
> | Conditional LM (full) (reported in Table.1)  | 80.3  | 78.9  | -50.8  | 41.9 |
> | CVAE  | 80.9  | 74.8  | -50.2  | 43.6  |
> | Ours (reported in Table.1) | **96.3**  | **59.8**  | -51.3  | 39.1 |

---

> > ### Comment · Reviewer_nG9H · 2021-08-22
> > **Response addresses my concerns**
> >
> > The authors clarify part of my concerns and also provide some new results to support the response. I decide to upgrade the original score to 6.

---

### Official Review · Reviewer_BdYj · 2021-07-18

**Rating:** 7
**Confidence:** 2

**Summary:**

The paper introduces methods to integrate causality in the task of controllable text generation, specifically on attribute-conditioned generation and text attribute transfer. The goal is to eliminate biases from spurious correlations with confounders (e.g., jobs when controlling the gender of the text). They do this by introducing a latent confounder variable and an observed proxy variable. They trained their models using a VAE objective, as well as three counterfactual objectives to help mitigate model collapse.

**Limitations And Societal Impact:**

The paper does not have separate sections for limitations and societal impact.

**Main Review:**

Strengths
* Controlling text while removing biases and spurious correlations is one of the most important problems in controllable text generation. The method introduced in the paper is intuitive and the results show that it works better than competing baselines.

Weaknesses
* It would be very helpful to readers who are more familiar with NLG but less familiar with work on causality to provide definitions to terms on this area (e.g., outcome, treatment, confounders, and their relations).
* The ablation could have been more extensive. For example, what happens when only one of the three counterfactual objectives is used?


**Time Spent Reviewing:**

8

---

> ### Author Response · Authors · 2021-08-10
> **Response**
>
> Thank you for your positive feedback.
>
> &nbsp;
>
> **Definitions of causality terms**
>
> In the paper we included a Background section (section 2) to briefly introduce the concepts in causality. We’ll add more details to make the background clearer.
>
> &nbsp;
>
> **Ablation studies**
>
> As discussed in the paper (Lines.193-201), the first counterfactual (*cf*) objective concerning the target attribute (Eq.4) is indispensable as it avoids model collapse (i.e., completely ignoring the conditioning attribute, leading to no control over the generation). For ablation study, we’ve reported the results when only the first counterfactual objective is used while the second/third objectives are omitted (i.e., *Ours w/o cf-z/c*).
>
> Below we additionally report the results when only the second objective is omitted (i.e., *Ours w/o cf-z*) for more ablation analysis. (We omitted the fluency/diversity results as they are similar across the methods.) We can see that, compared to *Ours w/o cf-z/c*, the added counterfactual objective consistently reduces the bias. But it still falls behind our full method equipped with all three counterfactual objectives. The results thus demonstrate the effect of both the second and third counterfactual objectives. We’ll include more results and discussion in the revised version.
>
> &nbsp;
>
> * *Additional ablation results for attribute-conditional generation (Table.1) on Yelp*:
>
> | Methods   | Control accuracy  | Bias (the lower the better)  |
> |---|---|---|
> | Ablation: Ours w/o *cf-z/c* (reported in Table.1) | 91.1  | 89.2 |
> | Ablation: Ours w/o *cf-z*      | 93.5  | 72.3  |
> | Ours (reported in Table.1)      | **96.3**  | **59.8**  |
>
> &nbsp;
>
> * *Additional ablation results for text style transfer (Table.3) on Yelp*:
>
> | Methods   | Control accuracy  | Bias (the lower the better)  |  Preservation |
> |---|---|---|---|
> | Ablation: Ours w/o *cf-z/c* (reported in Table.3) | 75.0  | 67.8 |  36.3  |
> | Ablation: Ours w/o *cf-z*      | 75.3  |  64.4  |  37.9  |
> | Ours (reported in Table.3)      | **77.0**  | **61.4**  |  **42.3**  |

---

> > ### Comment · Reviewer_BdYj · 2021-08-23
> > **Thanks for the response.**
> >
> > I have read the response and I think they addressed my concerns well.

---

### Official Review · Reviewer_rETv · 2021-07-19

**Rating:** 6
**Confidence:** 4

**Summary:**

The authors propose a unified causal framework for controllable text generation tasks (attribute-conditional text generation and attribute text transfer). The authors propose a structural causal model (SCM) to formulate the two tasks as causal intervention and counterfactual reasoning in the context of causal latter, which thus provides a unified framework for studying controllable text generation.  The learning stage aims to integrate variational learning (VAE) and confounder disentanglement (and balancing).

For the evaluation, the authors reused Yelp and Bios datasets, and then added synthetic spurious correlations, creating challenging data scenarios.  They evaluate the proposed model with a few simple baseline methods in terms of control accuracy, bias, fluency, and diversity. To support the automatic evaluation of these metrics, they use pre-trained attribute classifiers. They also conduct human evaluations to show the effectiveness of their proposed methods.

Overall, the main contribution of the paper is a casual perspective for two main controllable text generation tasks via casual interventional inference and counterfactual reasoning. The method can be used to reduce toxic content (e.g., gender bias) in downstream applications that require automatic text generation.


**Limitations And Societal Impact:**

***Have the authors adequately addressed the limitations and potential negative societal impact of their work?***

This proposed method in this paper can be used to reduce the bias in automatic text generation tasks and thus can lead to a positive social impact. However, it is also possible that someone can use the method to generate even toxic text by controlling the attribute in a bad way.  In this aspect, I would suggest the authors add a simple paragraph in the final version to discuss the social impact from both aspects.

**Main Review:**

***Originality: Are the tasks or methods new? Is the work a novel combination of well-known techniques? (This can be valuable!) Is it clear how this work differs from previous contributions? Is related work adequately cited?***

The controllable text generation tasks,  particular causal framework, and learning methods are all existing techniques. The work is rather a novel combination of them and provides a novel and valuable perspective. The casualty-centric framework is adequately different from prior works in the same direction, most of which are also well cited.


***Quality: Is the submission technically sound? Are claims well supported (e.g., by theoretical analysis or experimental results)? Are the methods used appropriate? Is this a complete piece of work or work in progress? Are the authors careful and honest about evaluating both the strengths and weaknesses of their work?***

The proposed method itself is sound and complete. The evaluation section is rather inadequate. There are only two simple datasets and the attribute and confounding factors are very simple. The baseline methods for controlling attributes and debiasing are not compared. The authors only used a simple "Conditional LM" as the key baseline method. Therefore, the comparisons are not that convincing to me.



***Clarity: Is the submission clearly written? Is it well organized? (If not, please make constructive suggestions for improving its clarity.) Does it adequately inform the reader? (Note that a superbly written paper provides enough information for an expert reader to reproduce its results.)***

Most of the parts are clearly written, while I would suggest the authors add more concrete examples of attributes vs confounding factors in the method section. Maybe the authors can simply add a problem formulation section and use the cases in the dataset description paragraphs (e.g., sentiment vs categories). Another improvement of the presentation could be adding a figure for the overview architecture of the proposed method, i.e., connecting the GPT2 and the attribute classifier and so on to illustrate how the method works in terms of the implementation.


***Significance: Are the results important? Are others (researchers or practitioners) likely to use the ideas or build on them? Does the submission address a difficult task in a better way than previous work? Does it advance the state of the art in a demonstrable way? Does it provide unique data, unique conclusions about existing data, or a unique theoretical or experimental approach?***

As stated above, the empirical results in the paper are inadequate, although the reported numbers show that the proposed method is much better than a simple baseline. It is, however, not convincing enough that the proposed method is significantly better than more sophisticated baseline methods --- either recent conditional text generation methods or the methods to remove spurious correlation (for NLG and other tasks). The style transfer experiment does not use other popular datasets in the community, so it is hard to directly compare it in a broader context.

Also, the analysis of the results is also superficial and does not give any insights and implications why such a casual framework is significantly better. The experiment section is barely a report of the results. Adding more dedicated analysis with concrete examples would improve the paper a lot.

***Other weaknesses. ***

The proposed method (and the evaluation) is only limited to a single target attribute and a single confounding factor. What are the key challenges for extending the method (and the experiments) to multiple target attributes with multiple confounding factors? The authors should have explicitly discussed the limitation and the possible ways to address them as this is a key weakness of the paper.

Minor question: Would it be possible to apply the method to lexical-constrained text generation tasks such as CommonGen?

**Time Spent Reviewing:**

2

---

> ### Author Response · Authors · 2021-08-10
> **Response**
>
> Thank you for your insightful comments!
>
> &nbsp;
>
> **Comparison with more baseline methods**
>
> In the paper, we’ve discussed the limitations of the most recent conditional text generation and debiasing work (e.g., Lines.33-39, and in Related Work section). Those methods are often either designed for a particular attribute (especially “gender”), or relying on *a priori* debiased classifiers often not available in practical cases.
>
> For a more adequate comparison as suggested by the reviewer, we evaluate two additional baselines published most recently, including:
>   - Fair-reg [21], which debiases the training of a language model (LM) `p(text | attribute)` by applying a fairness regularizer based on a confounder classifier `p(confounder | text)`.
>   - GeDi [31], a conditional decoding method that controls a base language model (“base-LM”) by applying another conditional language model (“gedi-LM”) that reshapes the generation distribution. To apply it to our problem, we use the `p(text | attribute)` LM as the base-LM, and the `p(text | confounder)` LM as the gedi-LM. That is, we can control both the attribute and confounder of the generation, similar to the *“Conditional LM (full)”* baseline in our paper.
>
> For a fair comparison, we used the same model architectures and data as in our paper to train the above two baselines, and tuned key hyperparameters following the previous papers/official code (e.g., regularizar strengths). The results are shown below:
>
> &nbsp;
>
> * *Results on Yelp*:
>
> | Methods   | Control accuracy  | Bias (the lower the better)  |  Fluency  |  Diversity  |
> |---|---|---|---|---|
> | Fair-reg [21] | 78.7  | 68.1  | -53.7  | 40.2  |
> | GeDi [31]     | 80.9  | 74.3  | -83.2  | 41.7  |
> | Ours (reported in Table.1)      | **96.3**  | **59.8**  | -51.3  | 39.1  |
>
> &nbsp;
>
> * *Results on Bios*:
>
> | Methods   | Control accuracy  | Bias (the lower the better)  |  Fluency  |  Diversity  |
> |---|---|---|---|---|
> | Fair-reg [21] | 91.7  | 71.3  | -23.9  | 42.8  |
> | GeDi [31]     | 86.0  | 75.2  | -27.8  | 43.5  |
> | Ours (reported in Table.1)   | **99.2**  | **62.4**  | -32.0  | 40.6  |
>
>
>
> We can see that our approach achieves higher control accuracy and greatly reduced bias (the same conclusion in the paper). More specifically, Fair-reg, as discussed in the paper, by design relies on a high-accuracy unbiased confounder classifier to effectively regularize the LM, and thus has inferior performance in our practical setting where only a small subset of confounder labels are available for training the classifier (Line.78). GeDi falls short of effectively reducing the bias while retaining the control accuracy due to the trade-off between base-LM and gedi-LM during inference. We will add the results and more details in the revised version.
>
> &nbsp;
>
> **Single target attribute and single confounding factor**
>
> We’d like to first note that it is the most common setting in previous text generation debiasing work to focus on single target attribute and confounding factor [e.g., 21,37,64] (even further, many of the studies in this line focus specifically on the “gender” attribute [e.g., 10,63,65,66,82]).
>
> The proposed framework can easily accommodate multiple target attributes and confounding factors. Specifically, for multiple confounding factors, it suffices to set the variable `c` to be multi-dimensional to capture all confounding information. For multiple target attributes, the modeling is done by having multiple attribute variables `{a_1, a_2, ...}`, where each variable `a_k` has two edges `z -> a_k` and `a_k -> x` in the SCM. Each target attribute has an attribute classifier to form the counterfactual loss Eq.(4) for training. We’ll add the discussion in the revised version.
>
> To the minor question: With the above, we can adapt the method for unbiased lexical-constrained text generation (e.g., to remove the spurious correlation with gender when an occupation-related word is an lexical constraint) (one can replace the attribute classifier in the counterfactual loss Eq.(4) with any differentiable lexical-constraint satisfaction criteria). Thus it’d be interesting to study the application of the method to CommonGen, even though it is not designed to study the *unbiased* text generation problem.
>
> &nbsp;
>
> **Datasets**
>
> The two corpora we used are among the most popular large datasets in related research, and crucially, have both attribute and confounder labels for our problem setting. Specifically, BIOS is one of the largest biased text corpora with both gender and occupation labels, and is widely used in research of debiasing text understanding and generation. Similarly, YELP has both sentiment and category labels, which allows us to derive challenging datasets for our experiments. A majority of other popular text style transfer datasets were not designed for the debiasing problem, and contain only the target attribute labels (without confounder labels).
>
> &nbsp;
>
> **Writing**
>
> Thanks for the nice suggestions. We’ll add a problem formulation section with concrete examples to make the problem clear. We’ll also add a figure to illustrate the architecture of the proposed method.
>
> We discussed the intuition of our causal framework and its superiority over existing ML-based approaches in the method section (section 3). We’ll add more discussion and analysis to the experiment section to highlight the insights. The supplementary materials include generated examples. We’ll add them to the experiment section for illustration and analysis.
>
> We’ll add more social impact discussion.

---

> > ### Comment · Reviewer_rETv · 2021-08-23
> > **Thanks for the response.**
> >
> > I have read the response of the authors. The results for comparing with two more baseline methods make the evaluation in the submission stronger and more convincing, which is what the authors should have done in the initial submission. As for the limitation of single-factor, I know the selected simplified setting in the paper is indeed the most common choice, while in terms of novelty and advancing the community it's a bit disappointing not to explore multi-factor controlling. The authors haven't tried to address my concern about the societal impact in the rebuttal, though.
> >
> > A minor note: would the authors further explain the reason why both baseline methods have a higher diversity than the proposed method?
> >
> > Overall, I'd like to keep my score for this submission.

---

> > > ### Author Response · Authors · 2021-08-24
> > > **Response to new comments**
> > >
> > > The additional regularizations in training and inference (for debiasing the generation) tend to lower the diversity to some extent. This seems to be a general observation -- e.g., the diversity of the baselines *Fair-reg* (40.2) and *GeDi* (41.7) is slightly lower than that of the *conditional LM* (41.9, see Table.1). Our method has slightly lower diversity compared with the baselines, with much improved accuracy and reduced bias.
> > >
> > > We've briefly mentioned in the *Conclusion* section about the potential malicious use of controllable text generation for producing toxic content. We'll add more discussion.

---

### Author Response · Authors · 2021-08-10
**General response**


We thank all the reviewers for their insightful and encouraging comments. We’re encouraged by the reviewers’ appreciation that: **(a)** the proposed causal framework provides a novel and valuable perspective (R1) for controllable text generation; **(b)** the studied problem of removing biases in controllable generation is one of the most important problems (R2) and the problem setting is practical (R4); **(c)** the method is sound, complete (R1), and intuitive (R2); **(d)** the results show the method works well (R2, R4); and overall **(e)** the paper has ample novelty and significance (R1, R4), cites related work well (R1), and is clearly-written (R3).

We highlight that the **main contribution** of the paper is a novel unified causal framework for controllable text generation, which leads to the key advantage of reducing bias and spurious correlations in the training data using principled causality techniques. And thanks to the unified framework, the same techniques can be applied to debias different controllable generation tasks including attribute-conditional generation and text attribute transfer, which is not possible in previous methods. To the best of our knowledge, this is the first causal formulation for controllable text generation, which opens up many new opportunities of introducing principled causality tools for tackling bias, robustness, and other modeling problems in NLP.

*(Note: R1 - Reviewer rETv; R2 - Reviewer BdYj; R3 - Reviewer nG9H; R4 - Reviewer dfNU)*

---

### Decision · Program_Chairs · 2021-09-27

**Decision:**

Accept (Poster)

**Comment:**

The submission proposes a method for debiasing controllable text generation, allowing text to be generated that satisfies one desired attribute, without also introducing confounding attributes. The reviewers agree that this is an important problem, and that a causal perspective is valuable here. The major concerns raised by the reviewers focused on the evaluation, however the authors have provided additional experimental results in their response that significantly strengthen the conclusions. Therefore, I recommend acceptance.